# Are Absorbable Plates More Resistant to Infection Than Titanium Implants? An Experimental Pre-Clinical Trial in Rabbits

**DOI:** 10.3390/jfb14100498

**Published:** 2023-10-09

**Authors:** Dimitrios Kitridis, Panagiotis Savvidis, Angeliki Cheva, Apostolos Papalois, Panagiotis Givissis, Byron Chalidis

**Affiliations:** 11st Orthopaedic Department, School of Medicine, Faculty of Health Science, Aristotle University of Thessaloniki, 54124 Thessaloniki, Greece; dkitridis@gmail.com (D.K.); panosortho@hol.gr (P.S.); pgivissis@gmail.com (P.G.); 2Department of Pathology, School of Medicine, Faculty of Health Science, Aristotle University of Thessaloniki, 54124 Thessaloniki, Greece; antacheva@yahoo.gr; 3Experimental Research Center ELPEN, 19009 Athens, Greece; apapalois@elpen.gr

**Keywords:** absorbable implants, PLLA, infection, titanium, rabbits

## Abstract

*Background*: Infection of orthopaedic implants after internal fixation of bone fractures remains a major complication with occasionally devastating consequences. Recent studies have reported that the use of absorbable materials, instead of metallic ones, may lead to a lower incidence of postoperative infection. In this experimental pre-clinical animal study, we compared the infection rate between absorbable implants consisting of copolymers composed from trimethylene carbonate, L-polylactic acid, and D, L-polylactic acid monomers, and titanium implants after the inoculation of a pathogenic microorganism. *Material and Methods*: We used an experimental implant-related infection model in rabbits. Sixty animals were randomly and equally divided into two groups. In all animals, the right femur was exposed via a lateral approach and a 2.5 mm two-hole titanium plate with screws (Group A), or a two-hole absorbable plate and screws (Group B), were applied in the femoral shaft. Afterwards, the implant surface was inoculated with Pseudomonas Aeruginosa at a concentration of 2 × 10^8^ CFU/mL. The primary outcome was the comparison of the incidence of developed infection between the two groups. The wound condition was monitored on a daily basis and radiographies were obtained at 12 weeks postoperatively. Infection-related laboratory markers (white blood cell count, erythrocyte sedimentation rate, and C-reactive protein values) were assessed at 3, 6, and 16 weeks postoperatively. Histologic analysis and cultures of tissue samples were also performed to evaluate the presence of infection. *Results:* Clinical and laboratory signs of infection were evident in 11 rabbits in Group A (36.7%), and 4 in Group B (13.3%). The difference between the groups was statistically significant (*p* = 0.04). Five animals in Group B (16.7%) had clinical and histologic signs of a foreign-body reaction with significantly elevated CRP and ESR values but no simultaneous presence of infection was identified (*p* = 0.04). Bone remodelling with thickening of the periosteum and surrounding sclerosis was demonstrated radiologically in animals developing infection or foreign-body reactions. *Conclusions:* Absorbable plates and screws show lower susceptibility to infection compared to titanium ones. However, their application is associated with foreign-body reaction and the potential need for a second surgical intervention.

## 1. Introduction

Infection remains a major complication after operative treatment of fracture fixation with metallic plates and screws [1,2,3,4,5]. These implant-associated infections usually lead to prolonged treatment times, additional surgical procedures, and poor functional outcomes [5]. Further complicating the issue, the formation of biofilm on orthopaedic implants increases antimicrobial resistance and tolerance to a broad range of antibiotics [6]. Biofilm can be defined as a structured microbial community of cells that are attached to a substratum and embedded in a self-produced matrix of extracellular polymeric substances [6]. The formation and maturation of biofilm on implant surfaces reduces the antibiotic action, impedes the phagocytosis and killing of bacteria by neutrophils, and increases the colonization of germs [1,2,3,7]. The bacterial biofilm thus encourages the persistence of infection, which could lead to implant failure, non-union, and osteomyelitis [5]. Infective non-union is a challenging and difficult condition to treat with often devastating consequences for patients’ health and quality of life [3]. Therefore, effective preventive and therapeutic measures to reduce the risks of implant-associated infections are necessary. Improvements in implant manufacturing and properties, the application of less invasive and tissue sparing surgical techniques, and perioperative local and systemic antibiotic prophylaxis have been considered to decrease the likelihood of infection development [1,2,3,5,7].

Bacterial colonization and proliferation are complicated processes influenced by many factors, including bacterial properties, implanted material characteristics, and environmental conditions [8,9]. The choice of the material used can affect the susceptibility to local infection [10,11,12]. Titanium plates are more biocompatible and confer greater resistance to infection than stainless steel [13]. Arens et al. have investigated the predisposition to infection after a local bacterial challenge using dynamic compression plates made of either stainless steel or titanium in rabbit tibiae. Under otherwise identical experimental conditions, the rate of infection for steel plates (75%) was significantly higher than that for titanium plates (35%) (*p* < 0.05) [11]. Likewise, absorbable plates used in fractures, osteotomies, and joint fusions have been also associated with lower rates of infection compared to metallic implants [14,15,16,17].

During the past decades, new absorbable plates and screws have been introduced as a reliable alternative option for treating bone fractures and lesions [18]. In the hand and wrist, absorbable pins and screws were used successfully for the fixation of fractures [19,20], osteotomies [20], and fusions [21]. Moreover, several series of operatively fixed ankle fractures have shown similar clinical and functional results between absorbable and metallic implants [22,23]. Absorbable implants, theoretically, eliminate the need for a secondary procedure for material removal after fracture healing [24,25]. The main disadvantage of absorbable materials is that the degradation process after their introduction could lead to a foreign-body reaction requiring surgical debridement and further operative procedures [26,27]. This adverse event has been gradually decreased with the evolution of the materials used to manufacture absorbable implants. First-generation absorbable implants, which appeared in the 1990s, were manufactured mainly with polyglycolic acid. However, they were associated with high rates of inflammatory reactions [28]. Second-generation implants consisted of poly-L-lactic acid, and showed a considerable improvement regarding the foreign-body reactions [27]. Even better results were achieved with the current third-generation implants consisting of copolymers composed from trimethylene carbonate, L-polylactic acid, and D, L-polylactic acid monomers [29].

The application of absorbable implants offers obvious clinical advantages by avoiding the presence of permanent foreign material in the body, provided that they could guarantee secure and stable fixation for an adequate time period. Third-generation absorbable implants were designed to overcome the problems of rapidly diminishing strength by extending the degradation time period [29]. According to the experimental studies, they lose most of their strength within 18–36 weeks, and complete bio-resorption takes place within two to four years [30]. Biomechanical studies have demonstrated that the primary fixation rigidity achieved with absorbable pins, screws, and miniplates for small bone osteotomies and fractures is close to that obtained with metallic fixation devices [19,31].

The Inion OTPS^TM^ Biodegradable Plating System is a third-generation absorbable implant system consisting of copolymers composed from trimethylene carbonate, L-polylactic acid, and D, L-polylactic acid monomers. The aim of the current experimental animal trial was to compare the incidences of infection and local tissue reaction between conventional titanium and Inion OTPS^TM^ absorbable fracture fixation materials after the inoculation of the implant interface with a pathogenic microorganism.

## 2. Material and Methods

The species chosen for this in vivo study were white New Zealand rabbits > 2500 kg. The choice of the animal model was based on the fact that in terms of anatomy and physiology the rabbits are comparable to humans and have similar skeletal growth patterns [32]. Furthermore, their size and strength make them suitable for surgical operations [33]. The largest long bone, namely the femora, was selected for the study to sufficiently accommodate the fixation implants.

Sixty male white New Zealand rabbits were used for this study (‘A. Trompetas’ Rabbit Farms, Athens, Greece). All rabbits were raised under standard farm management conditions and were routinely vaccinated. Ordinary physical and laboratory examinations was performed before housing in the facility of the provider, ensuring that the animals were healthy and pathogen free. The experiment was conducted in the laboratory of the Experimental Research Center ELPEN, which held an official permission for animal experiments (Registration number: 1-Ε.Π-05-1). Approval for the experiment was obtained from the Institutional Animal Care and Use Committee (IACUC) of Veterinary Directorate of the Prefecture of Athens (Approval Number: 56/10.12.18), according to Greek legislation and in conformance with the Council Directive 160/81 of the European Union.

The animals were accommodated in individual metal cages, fed with a standard pellet diet, and had free access to fresh water. Their mean weight at the beginning of the study was 3.32 kg (SD: 0.34 kg). Standard housing conditions with a temperature range between 18 °C and 22 °C, relative humidity between 55% and 65%, and day/night alteration at 6 am with the lights on to 6 pm with the lights off were applied. The cages were cleaned twice a week, and two veterinarians assessed the housing conditions, as well as the physiological condition and health status of the animals daily.

The rabbits were randomized into two groups using a software-generated list (www.randomizer.org, accessed on 17 December 2018). In the first group, a titanium plate and screws were implanted at the middle third of the femur (Group A: *n* = 30). In the second group, an absorbable plate and screws were introduced under the same setting (Group B: *n* = 30). A suspension of Pseudomonas Aeruginosa was inoculated into the examined femur of all animals, according to the experimental model described by Kälicke et al. [34].

### 2.1. Bacterial Suspension

A strain of *P. Aeruginosa* was obtained from a patient with chronic osteomyelitis. It was resistant to bacteriolysis when tested with fresh serum of five normal individuals. After overnight growth, an inoculum suspension of 2 × 10^8^ CFU/mL was prepared, according to the model by Kanellakopoulou et al. [35].

*P. Aeruginosa* microorganism was selected because it was capable to develop biofilms on prosthetic surfaces [35,36,37]. Moreover, Norden et al. reported that *P. Aeruginosa* osteomyelitis in rabbits presented with pathological and radiological changes that closely resembled those described in humans, and suggested that the model could be used for investigations of bone infection pathogenesis and therapy [38]. Just as importantly, *P. Aeruginosa* was associated with less severe symptoms in rabbits compared to the more common pathogens of osteomyelitis such as Staphylococcus Aureus or Epidermidis with respect to mortality, clinical illness, and formation of sequestra [38]. Therefore, postoperative monitoring of the animals could be more effectively facilitated.

### 2.2. Inoculation Method

There are three described methods of inoculation of implant interface that reflect the mechanisms of infection in the clinical settings [39]:In vitro inoculation; inoculation of the implant before implantation [40].Implant site inoculation; a pre-determined concentration of the bacterium is directly inserted into the implant site, either prior or after the implantation procedure [41,42].Intravenous injection of the inoculum; a pre-determined concentration of the bacteria is injected to replicate hematogenous spread [43].

The intravenous injection of the inoculum method was ruled out because the concentrations required to induce implant-related infection were often very large, and only low rates of contamination could be achieved [39]. Of the two remaining models, we chose the implant site inoculation method. The direct inoculation of exogenous bacteria on the plate bed during the operation was the best option to simulate a clinically relevant contamination and trigger infection [34].

### 2.3. Implants

In Group A, we used standard phalangeal 2.3 mm two-hole titanium plates (Stryker, Portage, MI, USA) fixed with two bicortical titanium screws. In Group B, we introduced 2.5 mm absorbable plates consisting of copolymers composed from trimethylene carbonate, L-polylactic acid, and D, L-polylactic acid monomers, which were fixed with two bicortical screws of the same material (Inion OTPS^TM^ Biodegradable Mini Plating System, INION, Tampere, Finland).

### 2.4. Surgical Procedure

All rabbits were operated on by two surgeons (PS, PG). Initially, the animals were anesthetized by intramuscular injection of a mixture of ketamine 50 mg/kg and xylazine 5 mg/kg. After successful induction of anesthesia, the animals’ right hind limb was shaved from the hip to the ankle, and it was placed onto the operating table. The surgeon then scrubbed in the conventional manner and gowned and gloved as per normal operative procedures. After sterile cleansing, the limb was wrapped using aperture drapes in an aseptic fashion. Using a simple sterile set of surgical instruments, a 6 cm long skin incision was utilized at the lateral aspect of the right femoral mid-diaphysis of each animal. Careful layer by layer dissection of all soft tissues facilitated adequate exposure of the femoral diaphysis.

After completing the approach, the metallic and absorbable implants were accordingly introduced. Each plate was placed in direct contact with the bone and stabilized with two cortex screws after bicortical drilling of the bone.

Subsequently, the experimental model of implant-related infection was established according to the model described by Kälicke et al. [34]. A sterile 18-gauge puncture needle was inserted over the surface of the plate and exited above the proximal wound margin. The wound closure took place in two layers; the deep fascia was closed with 2/0 polyglactin absorbable sutures and the skin with 2/0 silk non-absorbable interrupted stiches (Ethicon, Raritan, NJ, USA). After skin closure, 0.5 mL of bacterial inoculum suspension was inserted directly into the plate bed through the needle, followed by 0.1 mL of sterile saline to ensure complete entry of the suspension material. The needle was then withdrawn, and the wound was dabbed with a povidone iodine-soaked swab. Sterile dressings and elastic bandages were applied.

### 2.5. Postoperative Care

One preoperative antibiotic dose of subcutaneous enrofloxacin 10 mg/kg and oral carprofen 5 mg/kg/24 h for pain control during the first two days were administered. Acetaminophen suppositories were also used for one week to reduce animal pain and suffering thereafter. Suture removal took place at two weeks, under sedation, with intramuscular ketamine.

### 2.6. Clinical Assessment

The animals were monitored for a time period of 16 weeks. The wounds were examined on a daily basis and any local or systemic complications were recorded in a blinded manner. Wound redness, swelling, and drainage were assessed and recorded without knowledge of the experimental condition of any given animal.

### 2.7. Radiographic Assessment

Radiographs were obtained after 12 weeks under sedation with intramuscular ketamine 10 mg/kg and medetomidine hydrochloride 0.2 mg/kg. The reversal of the sedation was accomplished with administration of atipamezole 0.5 mg/kg. Any bone reactions including cortical destruction, lytic lesions, sequestrum formation, and soft tissue extension were monitored and evaluated.

### 2.8. Laboratory Testing

Complete blood count (CBC), erythrocyte sedimentation rate (ESR), and C-reactive protein (CRP) were measured preoperatively and at 3 weeks, 6 weeks, and 16 weeks postoperatively.

### 2.9. Histology

At 16 weeks, the animals were sacrificed under general anesthesia by a lethal injection of phenobarbital. Peri-implant samples from bone, periosteum, and soft tissues were received under sterile conditions and sent for histopathology. The tissue fragments were processed using routine histological procedures and embedded in paraffin. Each paraffin block was sectioned and stained with hematoxylin and eosin. The specimens were evaluated by one observer (AC) without knowledge of the group of the animals.

### 2.10. Stains and Cultures

The implants as well as the tissue samples including the bone, periosteum, and soft tissues around the plates were sent for Gram and modified Gram stains and culture examination in the same blinded fashion. Quantitative microbiological evaluation was performed under standardized test conditions. The target criterion was regarded as met if there was proof of the inoculated *P. Aeruginosa* strain in the samples.

### 2.11. Primary Outcome

The primary outcome of the study was the comparison of infection rates between the two groups. Infection was defined in case of positive tissue cultures and/or positive Gram or modified Gram stains [44]. A foreign-body reaction was confirmed if specific clinical, laboratory, and histological criteria were met (Table 1).

### 2.12. Statistical Analysis

The rates of infection between groups were compared with the Chi-square test. The Shapiro–Wilk test was used to assess if the quantitative variables were normally distributed. The CRP and ESR mean values were compared with the independent samples Kruskal–Wallis and the Mann–Whitney tests because they were non-normally distributed. The level of significance was set at *p* < 0.05. All data were analyzed using SPSS (Statistical Package for Social Sciences) software version 29.

## 3. Results

There were no complications related to the operative procedure and administration of the implants. The operations lasted an average of 20 min, and the anesthesia time was extended to an average of 60 min due to the need for shaving, sterile draping, and waiting for the recovery of anesthesia. There were no cases of mortality within the follow-up period. All animals completed the study uneventfully and, therefore, were included in the descriptive statistical analysis.

### 3.1. Infection Rates

Eleven animals in Group A (36.7%) and four in Group B (13.3%) were diagnosed with infection (Table 2). The difference between the groups was statistically significant (Chi-square test, *p* = 0.04). All animals detected with infection had redness and swelling on the operation site as well as abscess or fistula formation, with pus effusion. Moreover, in all infective animals the pathogenic (*P. Aeruginosa*) bacteria were isolated not only from the implant surface but also from the surrounding bone and soft tissue samples. Isolated evidence of bacteria development only at the implant material or underlying bone or neighbouring soft tissues was not identified.

### 3.2. Foreign-Body Reactions

Foreign-body reaction to absorbable implants was observed in five animals of Group B (16.7%) and none in Group A (*p* = 0.04). The image of the surgical wound resembled that of infection, but the cultures and histological examination of all the received tissue samples did not confirm the simultaneous presence of infection.

### 3.3. Laboratory Testing

Preoperative CRP and ESR values were normal in both groups. However, the animals with infection and aseptic reactions had significantly elevated CRP and ESR levels at all follow-up time points when compared to healthy ones (Table 3 and Table 4). The increase in CRP and ESR values in infection and foreign-body reaction cases had similar trends and no statistical difference between these two conditions was noticed (*p* > 0.05 in all time points, Mann–Whitney test). Details regarding the changes in CRP and ESR values are illustrated in Figure 1 and Figure 2.

### 3.4. Radiographic Assessment

The main radiographic findings among the animals with infection were bone remodelling with thickening of the periosteum and surrounding sclerosis (Figure 3). No bone sequestra was noticed. Animals with clinical findings of foreign-body reaction had similar radiologic appearance. Ten healthy animals in Group B also had periosteal thickening, while the remaining healthy animals in both groups had normal radiographic appearance and no bone reaction (Figure 4).

### 3.5. Histology

Histologic examination of samples from animals without any wound reaction did not reveal any bone or soft tissue abnormality (Figure 5). On the contrary, and in case of infection, loss of normal bone architecture, cortical thickening, and endocortical fibrosis were evident (Figure 6). When foreign-body reaction was apparent, numerous polymeric particles birefringent under polarized light were surrounded by macrophages and giant cells. Disruption of normal architecture with cortical thickening and dysplastic bone marrow were also observed (Figure 7).

## 4. Discussion

Infection remains a major complication after implant fracture fixation due to the potential formation of bacterial biofilm at the implant surface [1,2,3,4,7]. The use of absorbable materials is a well-promising alternative way of fixation as they have the theoretical advantage of resorption and eliminate the need for a subsequent operation for implant removal [24,25]. Nevertheless, the degradation process may lead to a foreign-body reaction which requires surgical debridement and additional surgeries [26,27]. We found that under identical experimental conditions, the absorbable plates had higher resistance to infection compared to titanium implants (13.3% vs. 36.7%, respectively). However, a clinically significant foreign-body reaction was observed in almost one-sixth of animals after the application of absorbable implants during a 16-week period of time.

The use of absorbable materials is constantly increasing in orthopaedic surgery and according to the literature, they yield lower rates of infections compared to metallic implants [16,17,46]. Sinisaari et al. reported a decreased incidence of wound infection when absorbable implants were used for fracture fixation, arthrodeses, and bone osteotomies instead of metallic devices (4% vs. 9%, respectively) [16]. The same authors, in another study, found that the occurrence of wound infection was slightly higher after the application of metallic plates and screws compared to absorbable devices (4.1% vs. 3.2%, *p* = 0.3) [46]. The exact mechanism by which the absorbable implants could affect the infection risk is not yet known. It is considered that the degrading process of absorbable materials can activate inflammatory cells that reduce the risk of bacterial colonization [47]. Devereux et al. observed that while the polyglycolic acid composite mesh had no intrinsic bactericidal or bacteriostatic activity, antimicrobial function could be achieved from the functional activation of leukocytes [47]. The authors advocated that this occurred within the time frame in which bacteria, that had either been seeded or translocated, began to grow and multiply.

The degradation process of absorbable material may stimulate a significant adverse tissue response leading to a foreign-body reaction and subsequent fixation failure, surgical debridement, and implant removal. This phenomenon may eliminate the benefit of using absorbable implants in pathological bone procedures and should be always taken into consideration [14,15,16,17,28,45]. The type of implant, the kind of manufacturing process and sterilization method, and the site of implantation could affect the degradation of the implant and the resulting biological response [26]. The modern third-generation absorbable implants that were used in our experimental study promise a better mechanical and biological environment for fracture healing as well as minimal reaction to the surrounding tissues [26,30]. Although we observed a foreign-body reaction in almost 17% of cases after implantation of absorbable implants, no simultaneous development of infection was noted. Therefore, these reactions could not be considered a predisposing factor for germ colonization and biofilm creation.

Among the metallic implants, titanium plates are considered the best choice for the prevention of infection. In a rabbit experimental study, Arens et al. reported lower rates of infection for titanium plates compared to stainless steel plates (35% vs. 75%, *p* < 0.05) [11]. Moreover, and using the same animal rabbit model, Cordero et al. proved that porous-coated cobalt–chromium implants required bacterial concentrations that were 40 times smaller than those needed to infect implants with polished surfaces, and 15 times smaller than those required to infect porous-coated titanium implants [10].

Experimental implant-related infection was surgically induced into the plate bed on the femoral diaphysis of rabbits by using the model of Kälicke et al. [34]. It should be noted that in our study only a few of the animals were infected, probably due to the application of antibiotics, which decreased bacterial adhesion [48]. It seems that the induction of the infection is a complicated process influenced by many factors, including bacterial properties, material surface characteristics, and environmental factors [8]. More specifically, the temperature and pH values, the time of exposure, and the associated biofilm structure and performance may influence bacterial adhesion [49,50]. Furthermore, the presence of serum proteins (i.e., albumin, fibronectin, fibrinogen, laminin, denatured collagen, and others) may promote or inhibit bacterial adhesion by either binding to substrata and bacterial surface, or just by being present [8].

The main strength of our study is the identical in vivo experimental conditions for all animals. On the other hand, the study has several limitations. For each group of animals, a specific plate in terms of size, length, and number of screws was applied in a specific anatomic area (femur diaphysis). Therefore, the results could not be generalized in all absorbable implants (pins, screws, rods, and plates) as well as in all anatomic areas and bone fractures or lesions. Similarly, the implant behavior might be different if other germs were inoculated. Finally, the mechanical behavior and stability of the fixation devices were not evaluated. Poor mechanical environments may predispose sites to fracture non-union and subsequent development of surgical infection and re-operation [51].

## 5. Conclusions

We found lower susceptibility of absorbable implants to infection in an experimental animal model. The promising clinical message of the study should be weighed in with the possibility of a second intervention due to potential foreign-body reaction. More studies are necessary to clarify the relationship between the degradation process and the inhibition of infection, as well as the exact role of the absorbable implants in the treatment of bone fractures and lesions.

## Figures and Tables

**Figure 1 jfb-14-00498-f001:**
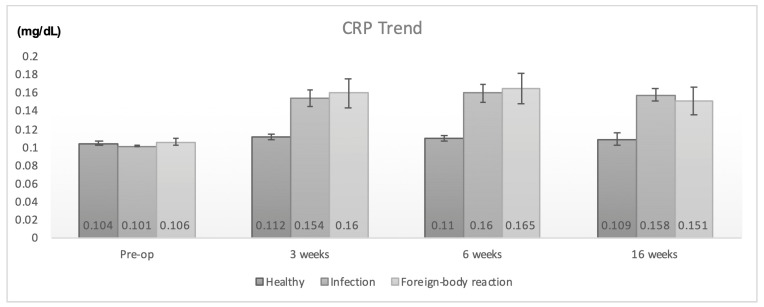
CRP values in healthy, infected, and animals with foreign-body reactions.

**Figure 2 jfb-14-00498-f002:**
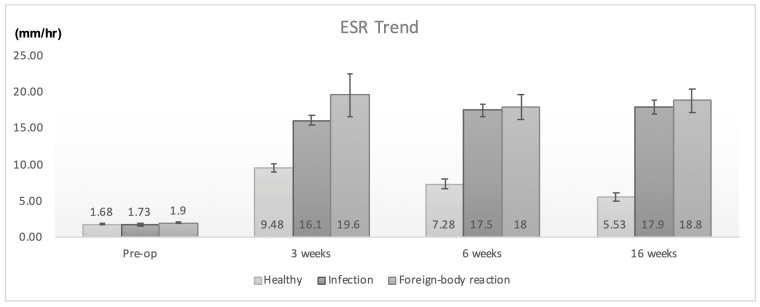
ESR values in healthy, infected, and animals with foreign-body reactions.

**Figure 3 jfb-14-00498-f003:**
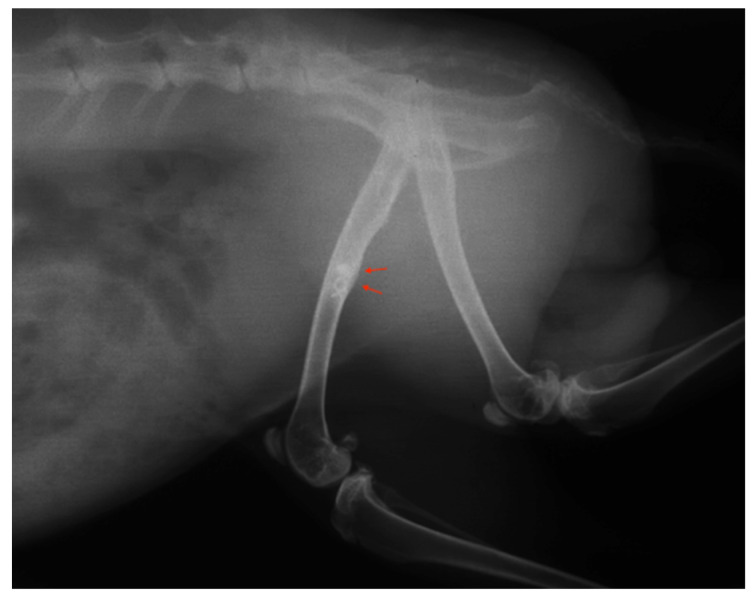
Radiograph of a rabbit with a titanium plate which developed an infection. A small periosteal reaction is apparent (red arrows).

**Figure 4 jfb-14-00498-f004:**
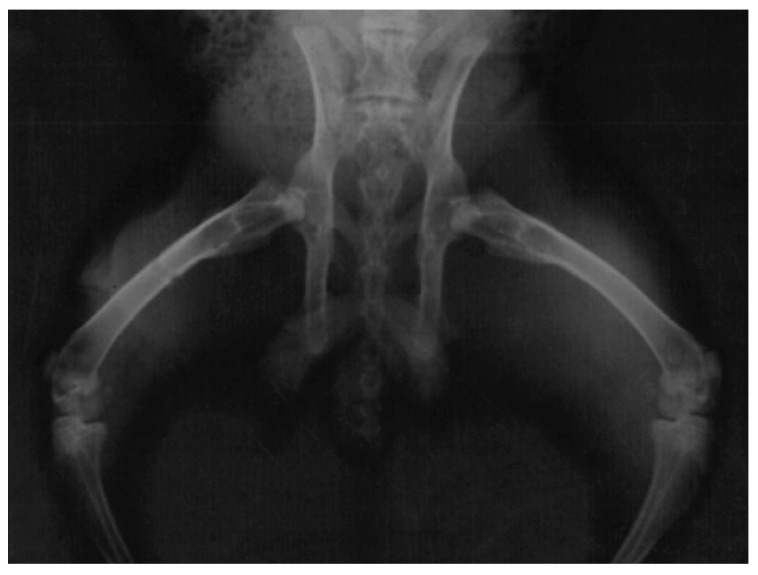
Radiograph of a healthy rabbit with an absorbable plate on the right femur and no bone reaction.

**Figure 5 jfb-14-00498-f005:**
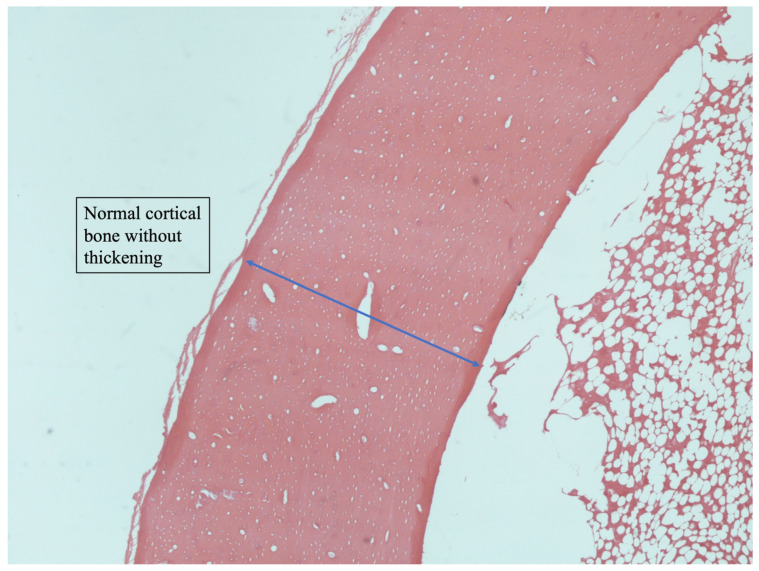
Histological image of cortical bone with normal architecture (blue arrow) (H + E stain, ×200).

**Figure 6 jfb-14-00498-f006:**
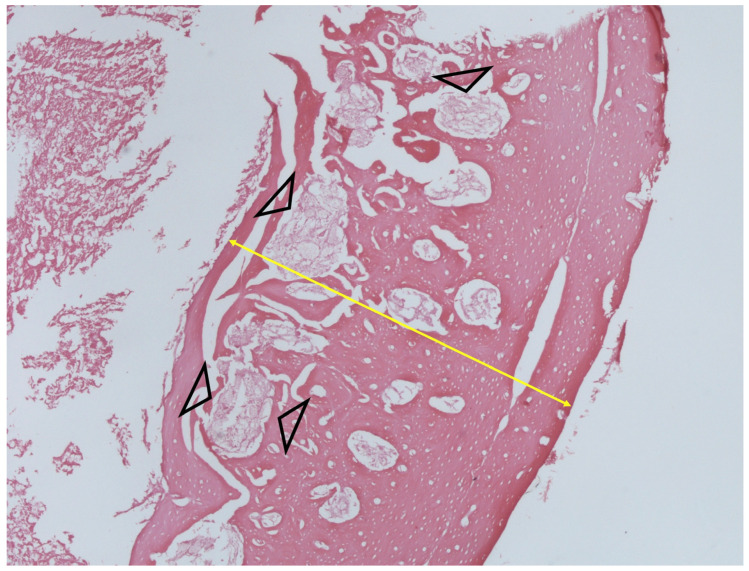
Histological image showing severe cortical thickening (yellow arrow) with diffuse endocortical fibrosis (arrow heads) in an infected bone (H + E stain, ×200).

**Figure 7 jfb-14-00498-f007:**
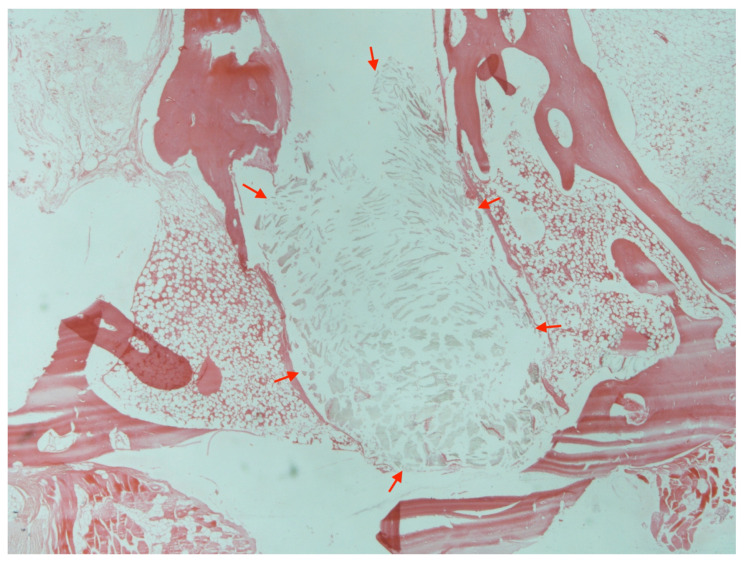
Histological image with particles of absorbable debris into the bone (red arrows) and loss of normal bone architecture in an animal with foreign-body reaction (H + E stain, ×200).

**Table 1 jfb-14-00498-t001:** Criteria for diagnosing foreign-body reaction [25,45].

Clinical	Wound Redness or Fluctuant Swelling or Wound Drainage
Laboratory	WBC > 15 × 10^9^/L, ESR > 3 mm/h, and CRP > 0.15 mg/dL (at least two out of three)
Histological	Nonspecific inflammatory reaction with numerous polymeric particles (birefringent under polarized light) phagocytosed by macrophages and giant cells, negative cultures, and Gram stain

**Table 2 jfb-14-00498-t002:** Rates of infection.

	Group A Titanium Implants	Group B Absorbable Implants	
*n*	30	30	
Infection (%)	11 (36.7%)	4 (13.3%)	*p* ^a^ = 0.04

^a^: Chi-square test.

**Table 3 jfb-14-00498-t003:** Mean (SD) CRP values (mg/dL).

	CRP (mg/dL)			
	Pre-op	3 weeks	6 weeks	16 weeks
Healthy animals (*n* = 40)	0.104 (0.016)	0.112 (0.019)	0.110 (0.019)	0.109 (0.020)
Infection (*n* = 15)	0.101 (0.004)	0.154 (0.036)	0.160 (0.038)	0.158 (0.026)
Foreign-body reaction (*n* = 5)	0.106 (0.009)	0.160 (0.036)	0.165 (0.038)	0.151 (0.035)
	*p* = 0.18 ^a^	*p* < 0.01 ^a^	*p* < 0.01 ^a^	*p* < 0.01 ^a^

^a^: Kruskal–Wallis test.

**Table 4 jfb-14-00498-t004:** Mean (SD) ESR values (mm/h).

	ESR (mm/h)			
	Pre-op	3 weeks	6 weeks	16 weeks
Healthy animals (*n* = 40)	1.68 (0.62)	9.48 (3.19)	7.28 (4.42)	5.53 (3.8)
Infection (*n* = 15)	1.73 (0.78)	16.1 (2.85)	17.5 (3.34)	17.9 (3.48)
Foreign-body reaction (*n* = 5)	1.9 (0.22)	19.6 (6.73)	18 (3.81)	18.8 (3.49)
	*p* = 0.39 ^a^	*p* < 0.01 ^a^	*p* < 0.01 ^a^	*p* < 0.01 ^a^

^a^: Kruskal–Wallis test.

## Data Availability

The data presented in this study are available in the main text.

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
