# Peer review of "Are Absorbable Plates More Resistant to Infection Than Titanium Implants? An Experimental Pre-Clinical Trial in Rabbits"

_jfb, 2023, doi:10.3390/jfb14100498_

Round 1

Reviewer 1 Report

As per my experience, this is not a clinical trial. It’s a pre-clinical animal study so it can’t be termed a randomized prospective trial. So please remove this term from the title as well as the text.

Provide the incidence rate of plate infection as per the literature in the introduction section.

Also, provide the mechanical properties comparison background of the used material with Titanium.

Results need to be present in sections such as clinical examination, Histopathological examination, etc.

Provide the error bar/ SD in graphs of Figures 1 and 2.

The quality of histological images are not so good, if possible replace them with other good representative images with well-labelled histological diagram.

Moderate editing of the English language is required

Author Response

Dear Editor,

We would like to thank you for accepting to reconsider our manuscript titled: “Are Absorbable Plates More Resistant to Infection than Titanium Implants? A Prospective Randomized Experimental Trial in Rabbits.” for publication in the Journal of Functional Biomaterials.

We would also like to thank the reviewers for their insightful comments. All points raised are addressed and the manuscript was revised according to their suggestions. All text changes in the manuscript have been highlighted. For reviewing purposes, the comments have been numbered and addressed one by one, and the text changes have been highlighted in the revised manuscript.

In more detail:

Reviewer #1:

COMMENT

RESPONSE

TEXT CHANGES

1. As per my experience, this is not a clinical trial. It’s a pre-clinical animal study so it can’t be termed a randomized prospective trial. So please remove this term from the title as well as the text.

Thank you for your comment. The title and text have been amended accordingly.

Updated title:

Are Absorbable Plates More Resistant to Infection than Titanium Implants? an Experimental Pre-clinical Trial in Rabbits.

Modifications in the Abstract/Backgroundsection:

“In this experimental pre-clinical animal study, we compared the infection rate between absorbable implants consisting of copolymers composed from trimethylene carbonate, L- polylactic acid, and D, L- polylactic acid monomers, and titanium implants after inoculation of a pathogenic microorganism.”

Modifications in the Introduction section:

“The aim of the current experimental animal trial was to compare the incidences of infection and local tissue reaction between conventional titanium and Inion OTPSTM absorbable fracture fixation materials after inoculation of the implants interface with a pathogenic microorganism.”

2. Provide the incidence rate of plate infection as per the literature in the introduction section.

Thank you for your comment. A wide range of infection rates after internal fixation of fractures have been reported in the literature. The level of bacterial contamination varies in relation of the type of the fracture and the condition of the soft tissues. Closed fractures present with lower rates of infection after internal fixation (up to 10%), while rates up to 50% have been reported for open fractures.

(1. Bonnevialle P, et al. Early surgical site infection in adult appendicular skeleton trauma surgery: a multicenter prospective series. Orthop Traumatol Surg Res 2012

2. Berbari E, et al. The Mayo prosthetic joint infection risk score: implication for surgical site infection reporting and risk stratification. Infect Control Hosp Epidemiol 2012

3. Oliveira P, et al. The incidence and microbiological profile of surgical site infections following internal fixation of closed and open fractures. Rev Bras Ortop 2016)

Apart from the fracture characteristics, the type of metal used seems to be important in the tissue reaction and bacterial adhesion. (Petty W, et al. The influence of skeletal implants on incidence of infection: experiments in a canine model. J Bone Joint Surg Am 1985). More specifically, application of titanium plates is complicated with infection in 0.8% to 29% of cases. (Arens S, et al. Infection after open reduction and internal fixation with dynamic compression plates - Clinical and experimental data. Injury 1996)

Arens et al. have conducted a prospective, randomized clinical study comparing 154 cases of stainless steel plates with 127 cases of titanium plates and the difference in the infection

rates showed no statistical significance, although a tendency in favor of titanium implants was apparent (6.3% vs 9.7% infection rates). The authors acknowledged several confounding factors in their study and emphasized on the need of in vivo experimental studies under controlled standardized

conditions. (Arens S, et al. Infection after open reduction and internal fixation with dynamic compression plates - Clinical and experimental data. Injury 1996). To further elaborate their results, Arens et al. have investigated susceptibility to infection after a local bacterial challenge using dynamic compression plates of either stainless steel or titanium in rabbit tibiae. Under otherwise identical experimental conditions the rate of infection for steel plates (75%) was significantly higher than that for titanium plates (35%) (p < 0.05).

(Arens S, et al. Influence of materials for fixation implants on local infection. J Bone Joint Surg Br 1996)

New additions to the Introduction section:

“Bacterial colonization and proliferation are complicated processes influenced by many factors, including bacterial properties, implanted material characteristics, and environmental conditions [8,9]. The choice of the material used can affect the susceptibility to local infection [10–12]. Titanium plates are more biocompatible and confer greater resistance to infection than stainless steel [13]. Arens et al. have investigated the predisposition to infection after a local bacterial challenge using dynamic compression plates made of either stainless steel or titanium in rabbit tibiae. Under otherwise identical experimental conditions the rate of infection for steel plates (75%) was significantly higher than that for titanium plates (35%) (p < 0.05) [11].”

3. Also, provide the mechanical properties comparison background of the used material with Titanium.

This is an interesting comment. The Inion OTPSTM products gradually lose most of their strength within 18-36 weeks, which is adequate for fracture union. Bioresorption takes place within two to four years. (Losken H, et al. Biodegradation of Inion Fast-Absorbing Biodegradable Plates and Screws. Journal of Craniofacial Surgery 2008)

Biomechanical studies have demonstrated that the primary fixation rigidity achieved with SR pins, screws and miniplates for small bone osteotomies is close to that obtained with metallic fixation devices (Waris E, et al. Bioabsorbable miniplating versus metallic fixation for metacarpal fractures. Clin Orthop 2003; Waris E, et al. Self-reinforced bioabsorbable versus metallic fixation systems for metacarpal and phalangeal fractures. A biomechanical study. J Hand Surg 2002). In more detail, in an oblique metacarpal osteotomy model, the 1.5-mm, self-reinforced, poly-L-lactide pins provided fixation rigidity comparable with 1.5-mm K-wires in dorsal and palmar apex bending, whereas in lateral apex bending and in torsion the rigidity was equal to that of 1.25-mm K-wires. The 2.0-mm, self-reinforced, poly-L/DL-lactide 70/30 screws provided rigidity comparable with that of 1.5-mm K-wires in all testing modes. The absorbable plate considerably enhanced the bending stabilities of the fixation system, but a single interfragmentary screw provided only limited rotational rigidity. These results show that absorbable implants adequate fixation stability for small bone fixation can be achieved. (Waris E, et al. Self-reinforced bioabsorbable versus metallic fixation systems for metacarpal and phalangeal fractures. A biomechanical study. J Hand Surg 2002)

New additions in the Introduction section:

“The application of absorbable implants offers obvious clinical advantages by avoiding the presence of permanent foreign material in the body, provided that they could guarantee secure and stable fixation for an adequate time period. Third-generation absorbable implants were designed to overcome the problems of rapidly diminishing strength by extending the degradation time period [29]. According to the experimental studies, they lose most of their strength within 18-36 weeks, and complete bioresorption takes place at two to four years [30]. Biomechanical studies have demonstrated that the primary fixation rigidity achieved with absorbable pins, screws and miniplates for small bone osteotomies and fractures is close to that obtained with metallic fixation devices [19,31].”

4. Results need to be present in sections such as clinical examination, Histopathological examination, etc.

Thank you for your comment. The Resultssection has been amended accordingly.

New sections of the Resultssection:

3.1  Infection rates

3.2  Foreign-body reactions

3.3  Laboratory testing

3.4  Radiographic assessment

3.5  Histology

5. Provide the error bar/ SD in graphs of Figures 1 and 2.

Thank you for your comment. The charts have been updated and they include error bars.

New versions of Figures 1 and 2 have been provided.

6. The quality of histological images are not so good, if possible replace them with other good representative images with well-labelled histological diagram.

Thank you for your comment. The figures have been enhanced and updated. New labels and arrows have been added, and the figure legends have been edited accordingly.

Figures 5, 6, and 7 and their corresponding legends have been updated.

Reviewer 2 Report

In their manuscript, the authors describe a possible link between resistance to Pseudomonas aeruginosa infection and the use of a bioresorbable implant. In an animal study of 60 rabbits, they showed that infection rates were lower when absorbable implants were used instead of titanium implants. However, the resorbable implants caused a foreign body reaction in some cases. I think the paper is suitable for publication in JFB.

I just have one small comment and one larger question out of interest that could possibly be included in the introduction as well.

Comment:

The manufacturer says:  “The Inion OTPS™ implants are made of biodegradable copolymers composed from L-Lactide, D-Lactide and TMC monomers.” I would add the fact that the implant is made of a copolymer to avoid confusion especially in “2.3. implants”(line 148-153). When I first read it, I wasn't sure if it was three different implants or maybe a PDLLA core and a TMC outer shell or something. And TMC is trimethylene carbonate.

And the question:

How often are resorbable implants used in clinical practice and what is the experience? And what are typical areas of application?

Author Response

Dear Editor,

We would like to thank you for accepting to reconsider our manuscript titled: “Are Absorbable Plates More Resistant to Infection than Titanium Implants? A Prospective Randomized Experimental Trial in Rabbits.” for publication in the Journal of Functional Biomaterials.

We would also like to thank the reviewers for their insightful comments. All points raised are addressed and the manuscript was revised according to their suggestions. All text changes in the manuscript have been highlighted. For reviewing purposes, the comments have been numbered and addressed one by one, and the text changes have been highlighted in the revised manuscript.

In more detail:

Reviewer #2:

COMMENT

RESPONSE

TEXT CHANGES

1. The manufacturer says: “The Inion OTPS™ implants are made of biodegradable copolymers composed from L-Lactide, D-Lactide and TMC monomers.” I would add the fact that the implant is made of a copolymer to avoid confusion especially in “2.3. implants” (line 148-153). When I first read it, I wasn't sure if it was three different implants or maybe a PDLLA core and a TMC outer shell or something. And TMC is trimethylene carbonate.

This is a fair point and thank you for your comment. We have amended the text accordingly.

Modifications to the Introduction section:

“Even better results were achieved with the current third-generation implants consisting of copolymers composed from trimethylene carbonate, L- polylactic acid, and D, L- polylactic acid monomers [29].”

“The Inion OTPSTM Biodegradable Plating System is a third-generation absorbable implant system consisting of copolymers composed from trimethylene carbonate, L- polylactic acid, and D, L- polylactic acid monomers.”

Modifications to the 2.3 Implants section:

“In Group B, we introduced 2.5 mm absorbable plates consisting of copolymers composed from trimethylene carbonate, L- polylactic acid, and D, L- polylactic acid monomers, which were fixed with two bicortical screws of the same material (Inion OTPSTM Biodegradable Mini Plating System, INION, Tampere, Finland).”

Modifications to the Abstract/Background section:

“In this experimental pre-clinical animal study, we compared the infection rate between absorbable implants consisting of copolymers composed from trimethylene carbonate, L- polylactic acid, and D, L- polylactic acid monomers, and titanium implants after inoculation of a pathogenic microorganism.”

2. How often are resorbable implants used in clinical practice and what is the experience? And what are typical areas of application?

Thank you for your comment. Exploration of the use of synthetic absorbable polymers for surgical fixation began in the 1960s. Since then, absorbable fixation devices are increasingly used in orthopaedic and trauma surgery. (Waris E, et al. Bioabsorbable fixation devices in

trauma and bone surgery: current clinical standing. Expert Rev of Med Devices 2004) Such implants are available for stabilization of fractures, osteotomies, bone grafts and fusions, as well as for reattachment of ligaments, tendons, meniscal tears and other soft tissue structures.

Several series of ankle fractures fixed with absorbable implants have been published so far. The absorbable implants showed clinical and functional results that were similar to those of metallic implants. (Gaiarsa G, et al. Comparative study between osteosynthesis in conventional and bioabsorbable implants in ankle fractures. Acta Ortopedica Brasileira 2015; Kukk A, et al. A retrospective follow-up of ankle fracture patients treated with a biodegradable plate and screws. Foot and Ankle Surg 2009) However, metallic implants are still recommended in comminuted fractures or non-cooperative patients. (Pelto-Vasenius K, et al. Redisplacement after ankle osteosynthesis with absorbable implants. Arch Orthop Trauma Surg 1998; Kankare J, et al. Malleolar fractures in alcoholics treated with biodegradable internal fixation. 6/16 reoperations in a randomized study. Acta Orthop Scand 1995)

In the hand and wrist, absorbable pins and screws are used in the fixation of fractures (Waris E, et al. Bioabsorbable miniplating versus metallic fixation for metacarpal fractures. Clin Orthop 2003; Kumta SM, Leung PC. The technique and indications for the use of biodegradable implants in fractures of the hand. Tech Orthop 1998; Pelto-Vasenius K, et al. Absorbable pins in the treatment of hand fractures. Ann Chir Gynaecol 1996) and osteotomies (Waris E, et al. Use of bioabsorbable osteofixation devices in the hand. J Hand Surg Br 2004), and to stabilize fusions (Voutilainen N, et al. Arthrodesis of the wrist with bioabsorbable fixation in patients with rheumatoid arthritis. J Hand Surg 2002). Biomechanical studies have demonstrated that the primary fixation rigidity achieved with SR pins, screws and miniplates for small bone osteotomies is close to that obtained with metallic fixation devices (Waris E, et al. Bioabsorbable miniplating versus metallic fixation for metacarpal fractures. Clin Orthop 2003; Waris E, et al. Self-reinforced bioabsorbable versus metallic fixation systems for metacarpal and phalangeal fractures. A biomechanical study. J Hand Surg 2002).

The use of absorbable implants has also increased in joint and arthroscopic surgery. The main indications of absorbable implants in the knee joint are cruciate ligament surgery and meniscal refixation. Absorbable interference screws, wedges and transfixation devices have been used for reconsruction of anterior and posterior cruciate ligaments. (Kousa P, et al. Fixation strength of a biodegradable screw in anterior cruciate ligament reconstruction. J Bone Joint Surg 1995; Kousa P, et al. Initial fixation strength of bioabsorbable and titanium interference screws in anterior cruciate ligament reconstruction. Biomechanical evaluation by single cycle and cyclic loading. Am J Sports Med 2001; McGuire DA, et al. Bioabsorbable interference screws for graft fixation in anterior cruciate ligament reconstruction. Arthroscopy 1999)

Moreover, absorbable devices are used in the shoulder joint for soft tissue refixation and the treatment of shoulder instability (i.e. suture anchors, tacks, nails and wires). (Barber FA, et al. Sutures and suture anchors: update 2003. Arthroscopy 2003)

New additions in the Introduction section:

“During the past decades, new absorbable plates and screws have been introduced as a reliable alternative option for treating bone fractures and lesions [18]. In the hand and wrist, absorbable pins and screws were used successfully for fixation of fractures [19,20], osteotomies [20], and fusions [21]. Moreover, several series of operatively fixed ankle fractures have shown similar clinical and functional results between absorbable and metallic implants [22,23]”

“The application of absorbable implants offers obvious clinical advantages by avoiding the presence of permanent foreign material in the body, provided that they could guarantee secure and stable fixation for an adequate time period. Third-generation absorbable implants were designed to overcome the problems of rapidly diminishing strength by extending the degradation time period [29]. According to the experimental studies, they lose most of their strength within 18-36 weeks, and complete bioresorption takes place at two to four years [30]. Biomechanical studies have demonstrated that the primary fixation rigidity achieved with absorbable pins, screws and miniplates for small bone osteotomies and fractures is close to that obtained with metallic fixation devices [19,31].”